# “I Was Having My Midlife Fat Crisis”: Exploring the Experiences and Preferences of Home-Based Exercise Programmes for Adults Living with Overweight and Obesity

**DOI:** 10.3390/ijerph191912831

**Published:** 2022-10-07

**Authors:** Sofie Power, Nikita Rowley, Michael Duncan, David Broom

**Affiliations:** Institute of Health and Wellbeing, Coventry University, Coventry CV1 2DS, UK

**Keywords:** home-based exercise, obesity, overweight, semi-structured interviews

## Abstract

The involvement of people with lived experience in the design of exercise programmes is more likely to lead to a more needs-sensitive and population-specific intervention. There is limited evidence of the integration of people with lived experience, particularly regarding home-based exercise programmes for adults living with overweight and obesity, despite this being a population that would significantly benefit from a suitably tailored programme. Semi-structured interviews were virtually conducted to explore 20 participants’ experiences of exercising at home and their preferences for the design of future home-based exercise programmes. Codes were generated through thematic analysis, highlighting considerations such as comfort within a home-based environment, a desire for social connection, and the integration of technology. Four corresponding themes were generated, encapsulating participants’ choice reasoning for home-based exercise, difficulties of engaging in home-based exercise, undertaking and adhering to home-based exercise, and factors that constitute the perfect programme. Although the involvement of people with lived experience in research can be time-consuming, this process is fundamental to the design of an effective and efficacious programme. These findings will inform the design and development of a home-based exercise programme for adults living with overweight and obesity.

## 1. Introduction

Many home-based exercise programmes that have included (but are not restricted to) adults living with overweight and obesity have utilised established programmes that are neither specifically designed by, nor tailored to, this population group. Subsequently, they do not fully and accurately represent or consider the variables that may impact engagement in home-based exercise programmes. Prescribing a non-condition-specific exercise programme to population groups with health conditions and associated comorbidities can have detrimental effects on participants’ overall health [1]. It can also influence programme attendance and behaviour change [2], and does not accommodate for the increased programme design considerations that are greater than would be required for people classified as healthy [3]. Involving members of specific population groups, through opportunities such as Patient and Public Involvement (PPI), interviews, and focus groups will result in the development of more needs-specific and population-sensitive programmes [4].

Despite evidence promoting the value of involving individuals with lived experience in the design and production of scientific research interventions and programmes [5], there is limited evidence of practical application by researchers and professionals, particularly when designing and implementing home-based exercise programmes. As a research area that primarily investigates the effects of interventions on the health of specific population groups, where the involvement of affected individuals may improve their health, it would be considered paramount to include these groups within the design phase of the programme.

People living with overweight and obesity are more at risk of developing non-communicable diseases and experiencing adverse health events, such as cardiovascular disease [6] and some cancers [7], which can be reduced by increasing physical activity and reducing sedentary behaviour. Therefore, exercise programmes, as a mode of physical activity, should be carefully considered and prescribed to ensure safety whilst not compromising potential health benefits. This is a difficult balance to achieve, particularly in a home-based exercise programme where the level of supervision is typically almost non-existent.

Partially as a result of the recent COVID-19 pandemic, the demand for home-based exercise programmes across all population groups has increased, due to temporary closures of leisure and fitness facilities, and limitations on leaving places of residence for restricted periods of time. In addition, the continued rise of people living with obesity both in England [8] as well as globally [7], reinforces the need for preventative strategies such as increasing physical activity behaviour through exercise and reducing sedentary behaviour. At the time of writing, leisure facilities and community exercise opportunities have returned with minimal (if any) restrictions in England; however, there is still hesitancy by people to confidently return to pre-pandemic physical activity and exercise behaviours [9]. Furthermore, the benefits associated with home-based exercise programmes, including increased flexibility to suit lifestyles [10], reduced costs [11], and a potentially more comfortable environment without gym-associated weight stigma and feelings of shame [12], suggest that home-based exercise programmes may continue to be a popular option, particularly for adults living with overweight and obesity. To the best of the authors’ knowledge, there is a clear gap in the literature, particularly for adults living with overweight and obesity, a population group that would specifically benefit from a tailored home-based exercise programme, designed in collaboration with people with lived experience.

Our primary aim was to qualitatively explore the experiences of home-based exercise programmes for adults living with overweight and obesity. Secondly, we addressed a key barrier to continued participation and engagement by identifying preferences for the design of future home-based exercise programmes. Both of which were explored through semi-structured interviews.

## 2. Materials and Methods

Institutional ethics approval was provided by Coventry University Ethics Review Committee (Reference: P123487) with written and verbal informed consent provided by all participants before interview commencement.

### 2.1. Participants

Twenty adults (aged ≥ 18 years) were interviewed via Microsoft Teams (*n* = 18) or telephone call (*n* = 2). Individuals were recruited through social media, the National Institute for Health Research—People in Research website, and word of mouth, via snowball sampling. The deliberately broad inclusion criteria required participants to be aged ≥ 18 years and either living with or had previously lived with overweight (BMI ≥ 25 kg/m^2^) or obesity (BMI ≥ 30 kg/m^2^), or used exercise as a weight management mechanism.

### 2.2. Interview Guide Development

A semi-structured interview guide was developed in consultation with the research team. The guide was piloted with four individuals: one subject academic expert with expertise in physical activity, exercise, and health; two postgraduate research students studying physical activity and health; and one postgraduate research student in an unrelated subject area. This process tested the understanding, flow, and feasibility of questioning, as well as the approximate interview length. None of the specific interview questions or prompts required altering and were deemed fit for purpose. Minor adjustments were made to provide further information regarding the contribution of participant answers towards the development of future home-based exercise programmes.

Due to the wide discrepancy in the use of the term ‘home-based’ within the exercise and physical activity literature, a standardised definition was provided to participants at the start of the interview to ensure consistency and researcher-participant mutual understanding. The proposed definition defines home-based exercise as any exercise undertaken in the place of residence or the immediate vicinity, such as the garden and/or the driveway [13]. Participants were reminded before interview commencement that there were no correct or incorrect responses, and that the researcher was interested in their views and experiences—positive, negative, and indifferent.

### 2.3. Data Collection

All interviews were conducted by the lead researcher in a private environment to maintain participant confidentiality. Telephone interviews were recorded via voice recorder (Sony ICD-PX370), and interviews conducted via Microsoft Teams were recorded via both voice recorder and the online platform, which automatically generated transcripts. Following each interview, the lead researcher transcribed telephone interviews verbatim and checked the automatically generated transcripts for verbatim accuracy in comparison with the voice recording.

Completed transcripts were distributed to the participants for member checking to ensure that they were representative of the semi-structured conversation. Although member checking has been highlighted as an ineffective method for verification due to power dynamics, it can still provide an opportunity to develop further insight [14]. Considering the sensitive nature of the topic, the research team felt that it was important to provide this opportunity to the participants. All participants confirmed that the transcript was an accurate representation of the conversation, with minor amendments made to one transcript requiring clarification and typing inaccuracies, at which point all participants were assigned pseudonyms.

### 2.4. Data Analysis

Reflexive thematic analysis [15] was used to analyse the interview transcripts on NVivo 1.5. The research team took an experiential approach, guided by a critical realist ontology, generating codes and themes that give voice to participant’s meanings and experiences of home-based exercise.

The lead researcher consistently reflected on their personal experience and views of home-based exercise by completing a reflective journal after each interview, to reduce any potential influence whilst conducting the interviews and data analysis.

Analyst triangulation was undertaken with another member of the research team to highlight any selective perception or blind spots [16]. Discussion took place to agree on the generated codes and themes, in which both researchers’ thoughts were given equal value. Despite different codes, no conflicts in opinion that required mediation arose. Upon completing code and theme generation from all 20 transcripts, it was agreed that data saturation had been reached and no new information would be presented that could change the outcomes. Sufficient discussion and sense checking occurred with the remaining members of the research team to ensure that the themes and codes generated were representative of the dataset.

All numerical data are presented as the mean and standard deviation, unless otherwise stated.

## 3. Results

Nineteen of the twenty participants that undertook a semi-structured interview had previously undertaken home-based exercise in line with the earlier presented definition. Despite one participant having not undertaken home-based exercise, the interview was not terminated because it was deemed important to also understand reasons for non-engagement. The mean interview duration was 25 (±11) minutes.

### 3.1. Participant Demographics

One participant withheld information; therefore, demographics for nineteen participants are displayed in Table 1. The average age of participants was 42 (±17) years and all participants reported to be of White ethnic background.

Sixteen of the individuals that participated in the semi-structured interviews were categorised as having a Body Mass Index (BMI) ≥ 25 kg/m^2^. At the time of interviews, conducted between August and September 2021, there was at least one participant living within each BMI classification: healthy weight (*n* = 3), overweight (*n* = 6), obese (*n* = 9), and severely obese (*n* = 1). The mean BMI of all 19 participants was 31.19 (±6) kg/m^2^, classifying the average participant to be living with obesity at the time of interview.

### 3.2. Themes

After code generation, four key themes were identified and are displayed, alongside corresponding codes, as shown in Figure 1.

#### 3.2.1. Theme One: Why Choose Home-Based Exercise?

Theme one detailed participants’ reasons for undertaking specifically home-based exercise programmes over exercise away from the home. Not all of these reasons were solely tangible; it was deemed to be an accumulation of their external environment, internal beliefs, and recognition of the environmental differences that prompted choice and engagement in a specifically home-based exercise programme.

##### Awareness of the Benefits

Although not specific to home-based exercise, participants acknowledged that taking part in physical activity and exercise was psychologically and physiologically beneficial. They spoke of the benefits they had established an awareness of from a knowledge-based perspective. Tyler recognised and spoke about the value he placed on being active:

“*you know how important physical activity is into your daily life*”.

This was further echoed by Lauren, describing her knowledge on the health benefits and reasons for being more physically active:

“*it was contributing towards the weight loss that I needed ‘cause that’s what the doctor said*”.

Other participants spoke from individual experience, having personally recognised the authentic benefits of home-based programmes. Mary spoke about the benefits she noticed and how this transferred to other areas of her life:

“*I can see and feel a noticeable difference when I’m doing it to the rest of my daily life. I can notice a difference when I’m not more to the point. But it means that I can then go on and do more activities with other people in a way that I couldn’t do before*”.

The perceived positive changes that transpired created a positive feedback loop, reinforcing the benefits of being more active, encouraging participants to continue, generating further benefits.

##### Comfortable and Inclusive Environment

Participants spoke about home-based exercise being a more comfortable and inclusive environment, commenting that they did not feel confident exercising in a public setting and preferred their home environment. Heather spoke about accessing YouTube videos, having cancelled her gym membership, to increase exercise behaviour at home:

“*The two that I use are very inclusive… so it is quite welcoming not sort of you know clear off anyone who isn’t a size eight, doesn’t look fantastic in yoga pants and look absolutely gorgeous transitioning from warrior one to warrior two*”.

This was further highlighted by Daniel, who spoke about how home-based exercise was a solution to overcome the discomfort of exercising in a public environment:

“*That person comes here every day. He’s there with his mates, he’s doing his enormous weights. You know, here I am not doing that. And it may be silly…but it is one of those things that does affect how we do it. So it’s just finding a resolution which is the home-based exercise*”.

##### Enjoyment of the Exercise

One of the characteristics that also influenced participant’s choice was their enjoyment of the programme. Whether that was the specific type of exercise, the virtual social interaction or the endorphin release participants experienced. Victoria spoke about how her enjoyment of the exercise was paramount to continued engagement:

“*If you’re not enjoying it then what’s the point. Like you could do anything and not enjoy it but if you enjoy it then you’re more motivated to do it and you’re more likely to continue doing it and get into a habit*”.

##### Lower Activation Energy

Participants recognised that home-based exercise was logistically easier and more convenient. Christopher, amongst others, mentioned how not needing equipment made it easier:

“*That also takes away one of the barriers that I believe people might see, which is getting stuff ready. The preparation is minimal if you don’t need to prepare anything*”.

As echoed by Helen, the increased convenience of home-based exercise made it easier for participants to want to be more active:

“*I like the fact that I can do it quickly and easily and I don’t have a big activation energy to get to it…being able to pick up twenty minutes in the morning when I first get up…having that option would make me much more likely to do things versus well I can’t do an hour, so I’m not gonna do anything*”.

##### Social Support and Those around Them

One of the external influences saw participants describe how those around them motivated their engagement. Steven spoke about recognising their engagement in home-based exercise and how their physical health impacted exercise with his family:

“*So in the garden it may well be that the kids want to have a kick about, and then I might notice that at times I could find that hard work…any sort of activities that take place outside for me at my age are probably generated through children’s interest to do things*”.

William spoke specifically about the importance of his support network in using exercise as a tool for weight management, and how fundamental it was in this mechanism:

“*My weight has always been an issue to me and my management of it has always been an issue to me, but I’ve got a very strong support network. My wife is fantastic, she’s the one who’s got me out this latest rut, and undoubtedly I’m gonna go like that through life*”.

#### 3.2.2. Theme Two: Home-Based Can Be Harder

Theme two encapsulated both tangible and non-tangible challenges in the lives of participants that make it more difficult or prevent them from engaging in a specifically home-based exercise programme.

##### Can It Really Replace My Usual Exercise?

Participants highlighted that the home-based exercise they took part in was not a truly fulfilling alternative to their preferred method of non-home-based exercise, because of the lack of social interaction. Some participants even mentioned that they would rather not participate in home-based exercise at all, as highlighted by William:

“*I want that social, that team that competitiveness. It’s a sad way of looking at it but if it’s not going to give me that, I’d rather not do it*”.

Matthew also expressed a similar view:

“*I play football and badminton sometimes, rugby at the weekend sporadically and it’s all based on social interaction. I find it much more difficult to motivate myself to do activity by myself…the action of it to actually undertake it tends to be from a social point of view*”.

##### Don’t Know Where to Start

Participants, particularly those that disclosed additional health conditions and comorbidities, stated a lack of knowledge about where to start. Sharon commented that this was a particular concern, not wanting to hurt herself whilst doing something that she knew was meant to improve her health:

“*You know you can be over ambitious in what you’re doing and actually end up hurting yourself and I don’t want to do that. I want to stay as healthy as I can and knowledge of exercise is not very great*”.

##### Life Getting in the Way

Participants spoke about other commitments and responsibilities that often took precedence and impacted on their engagement. Participants mentioned influences such as fatigue from the work day, such as Christopher:

“*sometimes I just wanna get home and sleep…so sometimes it is quite difficult to get your head round to motivate yourself to do a workout after a long day*”.

Participants also found that household responsibilities were more obvious because they were spending more time at home, as summarised by Lauren who said:

“*Sometimes for me it’s just being at home. Oh I’ve got the washing up to do, I’ve got the cleaning to do*”.

Both of these commitments were deemed by participants as non-negotiable components of life, within the home environment, impacting their engagement in home-based exercise.

##### Physical Environment

The physical environment was highlighted as not being conducive to undertaking home-based exercise to their desired expectations. Often as a result of constraints such as a lack of suitable space and unsuitable equipment, as summarised by Emily:

“*We don’t have very much equipment, so I was doing it with like garden things like a rake and that wasn’t really suitable either. I prefer heavy weighted workouts and we didn’t have any heavy weights*”.

The lack of suitable environment was highlighted as a challenge, including a lack of exercise specific equipment. England saw a surge in demand for home-based exercise equipment, thus making alterations to the home-based environment even more challenging, as highlighted by Hannah:

“*I did look at it at the start of lockdown, but obviously prices just massively soared because everybody was doing it and you just couldn’t get anything ‘cause it was months away so we didn’t end up getting anything for any sort of exercise*”.

##### The Uncontrollable Barriers

A few of the challenges that participants also spoke about were person- and situation-specific, but deemed to be uncontrollable and unable to be planned for or tailored around. These were considerations such as injury, as highlighted by Thomas:

“*I injured my knee at work and so therefore I couldn’t do any kind of exercise for a week*”

Illness (both short- and long-term health conditions) was spoken about by Victoria:

“*if I’m like having an IBS attack the last thing I want to do is get up and start doing crunches and burpees*”.

Additionally, weather was spoken about by Daniel, where if the home-based alternative such as a treadmill was available, his seemingly uncontrollable challenge could be overcome:

“*one thing that really affects me is my inability to run in awful weather conditions and that throws off my entire day. It throws off sometimes my entire planned week of exercise*”.

#### 3.2.3. Theme Three: Undertaking and Adhering to Home-Based Exercise

Theme three highlighted potential changes to implement, both tangible and non-tangible, to overcome the previously identified challenges, detailing changes that were or could be implemented to make participation and continued engagement within a home-based exercise programme easier.

##### Community, Interactivity and Relatedness

As previously highlighted, participants presented a desire for the integration of a sense of community, interactivity, and relatedness, to not solely focus on the exercise, and to integrate a social aspect, as highlighted by Hannah:

“*So even though you’re at home, you’re still talking with other people, maybe doing like a group thing together, but doing it from your own homes would be quite nice, ‘cause then it’s not just the exercise, it’s seeing your friends as well*”.

The wish for community interaction was clear and would impact engagement, particularly in this statement by William, who said:

“*I need to find something that’s team related, and if I don’t have that, I’m not going to engage in it*”.

##### Integration of Technology

Participants consistently made reference to the integration of technology both for creating and accessing home-based programmes. As recognised by Heather:

“*I am conscious of the fact that that I’m lucky enough to have access to good Wi-Fi, iPad*”.

This was further echoed by Mary, whose engagement was positively impacted with the addition of technology:

“*Online you’ve gotta have a decent connection certainly at the moment, and I think if I wasn’t part of a class then I might do a week or two maybe, but it would wane you know, it would just disappear*”.

Technology also facilitated the desired social interaction by integrating technology into the programme design. However, the accessibility of existing technology-based programmes is limited, as highlighted by William, who said:

“*Peloton is home exercise for the rich. People can’t afford that. I can’t afford that*”.

Although this technology facilitates social interaction, there is a clear need for increased accessibility.

##### Lockdown Restrictions as a Result of COVID-19

The forced change in lifestyle as a result of the COVID-19 pandemic made participants more aware of the importance of physical activity, whilst providing a space and opportunity to prioritise this. Participants spoke about an increased awareness of their exercise behaviours within the restrictions of their current environment, as highlighted by Steven:

“*I suppose more recently, in lockdown you start to see the value of trying to be active wherever you can be, and grabbing moments to do it*”.

To further emphasise this, Lauren recognised and summarised the impact of the pandemic on physical activity behaviours:

“*At the minute, especially during COVID I think this has been actually one of the worst times for peoples fitness and health, or it’s been one of the best times*”.

##### More Knowledge and Guidance from a Trusted Source

Participants mentioned that the provision of knowledge and guidance from a reputable source would help make home-based exercise easier. Tyler spoke about the importance of a validated, trustworthy source:

“*I think if people know that it’s from a trusted source people are a lot more likely to use it*”.

The structure and details of this source varied between participants. Some participants felt that access to professional supervision would facilitate their participation, as presented by Lauren:

“*It would have been nicer to maybe have a personal trainer of some sort, who could have told me if my form was ok or not because even now I’m just going with the flow and hoping I’m doing it right*”.

Other participants felt that the provision of knowledge and resources would be helpful, as spoken about by Robert:

“*Access to more advice from the NHS for the moment because of my disability. The access to that information is probably more restricted than it would be under normal terms*”.

##### Source of Accountability and Motivation

A source of accountability and motivation was highlighted to make engagement in home-based exercise easier. Participants portrayed the idea that their engagement in the home-based exercise was lower and motivation was required to be higher when replicating a programme they would have previously been undertaken in an environment away from the home, as demonstrated by Victoria:

“*If it was something like burpees and I just couldn’t be arsed I just didn’t do it…but if obviously you’re at home there’s nobody to do that and if I tell them when I’m there that it’s making me feel sick they’ll get me to do something like squat jumps or something like that instead*”.

#### 3.2.4. Theme Four: What Makes the Perfect Programme?

Theme four encapsulated different characteristics presented throughout the interviews that would constitute the most ideal home-based exercise programme. The codes generated are not just tangible aspects, but collate participant’s previous experiences to create their ideal programme.

##### Flexibility in Accessibility

Being able to access their home-based programme whenever and wherever participants wished was key, and therefore fundamental in the development of a home-based programme. This flexibility reduced the challenges they previously experienced when trying to undertake exercise, as summarised by Emily:

“*If there were pre-recorded videos and stuff, I think that would probably work best for me ‘cause it means you can do it in your own time and no one has any excuses really*”.

The flexibility allowed participants to be active wherever in the home environment they felt most comfortable, as spoken about by Victoria:

“*If I did want to do yoga in my bedroom I could but if I wanted to do it downstairs I could or like if I’m away I can do it in my room*”.

##### Progress Monitoring and Biofeedback

The opportunity to monitor their progress and further justification for continued engagement was mentioned throughout the interviews, such as by Robert:

“*I would want to know effect essentially, is whether the exercise I was doing having a beneficial effect*”.

Participants also spoke about being surprisingly motivated by this, as demonstrated by Heather:

“*With the Fitbit it does reward you with little badges and achievements and you know the fact that you’ve walked the length of Italy. You’re thinking well I never knew I needed to know that but I like that. Those are definitely surprisingly encouraging*”.

William spoke about the impact of measuring his weight loss progress on mental wellbeing:

“*Don’t ever underestimate the difference in one or two numbers for someone who’s trying to lose weight because point five of a pound or point five of a kilogram taking you from 17 stone to 16 stone 13 and a half means absolutely everything to some people*”.

##### Relatability

There was a desire to relate other programme participants and the instructors, as presented by Tyler:

“*I need to feel that there’s people on the journey with me as well*”.

This was echoed by William, who noticed the physical difference between participants and instructors:

“*Everyone you see on a Les Mills video and Insanity video is shredded to hell. They all look absolutely incredible. We constantly talk in work about a person like me and being able to relate to someone*”.

Participants wanted the deliverers of the programme to look like them, and therefore to be able to relate to them. Without this, the programme was not conducive to engagement.

##### Variety Is Key

Participants spoke about the importance of variation for motivation and engagement. As highlighted by Emily, the opportunity to tailor the exercise programme to feeling was particularly important in ensuring that they at least did some exercise:

“*I think having a variety, at least if I was like, oh, I don’t want to do an aerobic session this week, I’ve still got the other two as options, so it’s not like I’m doing no exercise that week*”.

Participants also spoke about a need for variety in the exercise intensity, rather than focusing solely on the health benefits, as presented by Thomas:

“*Uh personally with me, I like the variation. So to go from light to moderate to intense, and then maybe back down again. You’re in those different zones and so you get a good feeling*”.

Virtual interactivity, social support, and a team environment

As previously highlighted, social connection was fundamental for participants to want to engage in home-based exercise. Tyler spoke how this would impact their ability to fully engage:

“*If I am going to exercise at home, like on a regular long-term basis, I need to feel like I’m a part of the community, and I need to feel that there’s people on the journey with me as well*”.

This was echoed by Hannah, when speaking about the opportunity to share achievements with people in similar situations:

“*I think it would also be fun if you could interact with others. So I know on some platforms you can like share your achievements, can’t you? So that would be quite nice if you’re kind of doing a group thing as well, so you can all share it and celebrate together anyone’s successes*”.

## 4. Discussion

### 4.1. Theme One: Why Choose Home-Based Exercise?

Participants spoke about characteristics they believed positively influenced their choice, and therefore, engagement within home-based exercise programmes. The combinations of these characteristics were recognised to encourage engagement in their chosen home-based exercise programme.

It was apparent that participants were aware, from a knowledge and experience grounding, that being physically active was good for their overall health and that knowing this encouraged them to engage in home-based exercise. The perceived positive changes as a result of engaging in home-based exercise created a positive feedback loop, with a positive affective response [17]. Reinforcing the benefits of increased activity encourages continuation, generating further benefits, creating a habit and long-term behaviour change.

The contrast in physical environment variables of a home-based programme in comparison with a programme away from the home were also highlighted. The home environment reduced the worry of others seeing them exercising, making mistakes or not knowing what to do, subsequently increasing feelings of inclusivity and comfort.

Enjoyment was another factor that became apparent when choosing home-based exercise, whether this was enjoyment of that specific type of exercise, the virtual social interaction, or the endorphins that participants felt. As highlighted by Victoria, enjoyment was key for participants to want to exercise at home [18] and to also make it habitual.

Participants recognised that undertaking home-based exercise was logistically easier and more convenient, reducing preparation time, travel time, expectation to appear publicly presentable, and set up equipment, whilst increasing flexibility to accommodate other obligations. The home-based environment removed barriers that would have previously stopped participants or reasoned for reduced engagement.

The social support network present in participant’s lives influenced their choice to engage in home-based exercise. Highlighting that home-based exercise can improve lives outside of solely intrinsic motivations, because the impact was bigger than just themselves. Participants recognised being physically active at home as both an opportunity to spend time with their family, alongside a medium to increase their health, helping them to be more active when spending time with those close to them. Participants, regardless of their demographic status, commented upon the role of their home social support network in motivating them to undertake home-based exercise to their best ability.

### 4.2. Theme Two: Home-Based Can Be Harder

Participants described elements in their lives that made it more challenging to engage in home-based exercise to their desired standard. It should be noted that not all of these barriers are specific to home-based exercise; they would still be present if the participants were trying to access exercise programmes externally.

Participants recognised the flexibility and convenience of home-based exercise; however, they also highlighted that it was not a fulfilling alternative to their desired exercise mode. The majority of this discrepancy originated from the decrease in social interaction, whilst also being reflected within the literature [19], and participants were unable to visualise how a home-based programme could replicate that.

Participants spoke about a lack of knowledge of where to start, eliciting feelings of being overwhelmed, and without knowing what was ‘right’, they were apprehensive about making mistakes. This paradox of wanting to engage in home-based exercise to improve health but not wanting to damage their health as a result of engagement was consistently raised by participants in the 45 years and plus age group. Fear of making mistakes or overdoing it prevented participants from fully engaging. When participants express a desire to be physically active, this should be something that is profited from.

Despite the majority of the participants in employment working from home at the time of interview, participants spoke about life getting in the way and the precedence of other commitments. Increased time spent at home was a catalyst to recognising other responsibilities they had to prioritise over home-based exercise. All the participants who mentioned a lack of time as a barrier to engagement immediately followed up with verbal recognition that they actually could make time, demonstrating a reduced capacity to prioritise home-based exercise over other commitments, rather than a lack of time as a barrier itself.

Although participants recognised that they were still able to be physically active at home, there was an acknowledgement of reduced physical space and lack of suitable equipment, both of which were not conducive to exercise at the same standard they could undertake away from the home and truly desired to engage in. Proposed changes to facilitate a larger space or more equipment were not quick or easy fixes, especially during the COVID-19 pandemic [20].

Some of the considerations impacting participants’ engagement were unable to be controlled, regardless of planning or changes to the home-based environment. Injury, illness, and weather conditions impacted participants to the extent that they commented upon them during interview. However, these considerations were not solely a result of the home-based environment, injury, and illness in particular impact exercise behaviour, regardless of the geographical location.

### 4.3. Theme Three: Undertaking and Adhering to Home-Based Exercise

The third theme encapsulated characteristics and changes that could be made to make engaging and participating in home-based exercise easier and potentially more enjoyable.

A sense of community, interactivity, and relatedness was important for participants, highlighting the role of social interaction in programme engagement. The integration of social connection reduces the focus on exercise [21], providing an opportunity for connecting with others in similar positions. Facilitating social connection would make the programme more representative of participant’s desires, of which was fundamental to facilitate initial engagement, not even continued engagement.

Participation in current programmes was facilitated by resources such as electronic tablets and stable internet connections. Participants recognised that technology to facilitate home-based exercise and social connection did exist; however, this was not widely accessible. Increased costs made exercise less accessible, at a time when public health should be a priority, thus increasing the risk of physical inactivity and widening of the socio-economic gap seen in population groups with low physical activity and high sedentary behaviour.

The lockdown restrictions as a result of COVID-19 forced significant changes in lifestyle. Participants become more aware of the importance of physical activity, whilst providing space and opportunity to prioritise engagement in home-based exercise. Although it was recognised that participants undertook this as a result of no alternatives, the restriction on community facilities and organised sport did facilitate participation within home-based exercise. It prompted an increased awareness of their current behaviours whilst providing space to consider improvements.

Further knowledge and guidance from a reputable source were desired. Particularly for those with additional health conditions, it was important to have access to digestible information that would provide reassurance for exercise engagement. The format of this guidance varied between participants; however, the need for it was clearly highlighted.

The presence of a source of accountability and motivation, whether that was a family member or fitness instructor, would facilitate their continued engagement [22]. Participants acknowledged that they could have put more effort in, attributing this to an absence of someone to push them or of motivation because they knew no one would notice if they did not fully engage.

### 4.4. Theme Four: What Makes the Perfect Programme?

Theme four encapsulated and provided direction for the development of future home-based programmes, providing considerations for designing programmes specifically for adults living with overweight and obesity.

Having access to their home-based programme whenever and wherever participants needed was key. Tailoring their engagement with the programme is therefore very important to consider in the design of future home-based programmes. If participants struggle to engage in exercise, making a programme malleable to their lifestyle and environment is fundamental to breaking that barrier. The implementation of this may require future programme designers to consider the format of home-based exercise programmes, the platforms on which they are delivered, and the timing of the sessions. For example, live virtual exercise sessions would restrict participants to a specific participation time. Ideally, programmes should provide participants access to videos at any time so they can tailor their programme engagement to their lifestyle.

Monitoring progress throughout the programme, often by biofeedback from wearable technology devices, was raised by participants throughout the interviews to monitor long- and short-term effects. These tools were motivating for the majority of participants, and provided a means by which they could recognise their achievements, which has also been highlighted within the literature [23]. Applying this to programme development would require considerations of feasibility and cost effectiveness; however, the inclusion of biofeedback would be beneficial for both researchers and participants.

Participants spoke of a desire to relate both to other participants engaging in the programme and, arguably more importantly, to visually relate to the health professional delivering the programme. Instructors in popular exercise programmes are often mesomorphic and meet the stereotypical ‘fit’ image, but participants were unable to relate to someone that did not look like them and found the exercise easy. The desire to visually relate may stem from the reduced communication of a virtual programme. This oversight may reduce motivation for participants by constantly exposing them to someone to whom they cannot visually relate, or to a seemingly unachievable goal. As a result, it is encouraged to include people with lived experience both in the design and the production of exercise programmes. Particularly programmes that utilise videos and visual aids, it would be important to ensure that these reflect the population group at which the programme is aimed.

Participants spoke about the importance of variety, whether this was the exercise type or the intensity, which made the programme more interesting and engaging. Participants wanted the option to tailor the programme to their mood, ability, and lifestyle changes. Whilst they recognised that increased engagement as a result of variety had beneficial effects on their physical health, it also increased their enjoyment. Implementing variety within home-based exercise programmes would require the integration, and promotion to participants, of exercise adaptations that consider the different needs of each participant. Although time-consuming, the inclusion of alternatives will contribute towards a more needs-sensitive programme.

The need for social connection and interactivity was fundamental if home-based exercise was the only option, presenting this as a non-negotiable element. Not only does social connection provide the opportunity to interact with others on a similar journey, but it is also a platform to share experiences and encouragement. Although not every participant mentioned that they would engage, the opportunity to choose was appreciated. This prompts the consideration of social connection opportunities and the platforms on which future home-based exercise programmes are delivered. Future programmes and delivery platforms should have a space where participants can socially connect, and virtually facilitate the interactions that people may have previously experienced during in-person exercise sessions.

### 4.5. Strengths and Limitations

Only one participant reported to not having undertaken home-based exercise; therefore, the data presented originated from a perspective of individuals that had at least some experience. No time restrictions for the exercise engagement period were set, and considering the COVID-19 restrictions over the past two years, it would be very difficult to find individuals, particularly with those living with overweight and obesity, that had not at least attempted some home-based exercise.

All participants that reported demographics were of a White ethnic background; therefore, these data are not representative of the experiences and preferences of people living with overweight and obesity of a Black, Asian, and Minority Ethnic (BAME) background. The recently published toolkit for increasing participation of BAME communities [24] should be implemented to target this limitation.

It was decided that virtual interviews were the most appropriate and feasible platform to collect data, aiding the ease of participation and increasingly being seen within the literature [25]. However, face-to-face interviews or a focus group setting could have produced richer data, although the sensitive nature of the topic may have limited the depth of conversation within a group setting. Considering the developments in technology to facilitate interviews and recording, it was deemed easier and safer without sacrificing data quality.

It would be important to recognise and consider the potential influence of the researcher’s positionality and individual preference regarding home-based exercise. If the researcher did not remain impartial throughout, the data collection and analysis would be representative of their own individual preferences rather than the participants. However, this was managed through the lead researcher’s journal, providing a space to maintain awareness and for the research team to continually challenge potential bias.

### 4.6. Direction for Future Research

Further research should be undertaken with members of BAME communities.

Specifically regarding practical application, themes and codes generated from these semi-structured interviews will inform the design of home-based exercise interventions for adults living with overweight and obesity, of which feedback will be received from people with lived experience as part of a PPI process. The generated programme and associated design process may then inform other researchers and practitioners within physical activity and health.

For home-based exercise intervention designs, it is important to consider how an element of social community can be introduced. Although community and centre-based environments are recognised as important, and the purpose of this study was not to encourage everyone to only exercise at home, it has further emphasised the need for population-specific home-based exercise programmes and prompts discussion for how researchers can best support this. It is an opportunity to make exercise and leading a healthier lifestyle accessible to all, especially those that may need it most.

More generally, the involvement of specific population groups when designing exercise programmes should be considered by researchers when creating a programme. Collaboration between people with lived experience and researchers with subject specific knowledge will only benefit the design of the programme for both participants and researchers. We recognise that involving members of the public in research is time-consuming and the suggestions are not always feasible to include within the developed programme; however, for specific population groups, it could be considered fundamental to the development of the research output and worth the initial time investment.

## 5. Conclusions

Home-based exercise programmes for adults living with overweight and obesity are an appropriate option for people wanting to be more active. However, programmes should be designed in collaboration with people with lived experience. This study explored the experiences and identified the preferences of home-based exercise programmes for adults living with overweight and obesity. However, the limited sample diversity prompts the need for further research into programme intervention development to be undertaken with members of BAME communities. Four key themes, with corresponding codes were generated, describing participant’s choice for home-based exercise, difficulties with home-based exercise, undertaking and adhering to home-based exercise, and the perfect programme. The results of these semi-structured interviews will inform the design of a home-based exercise programme sensitive to the specific needs of adults living with overweight and obesity.

## Figures and Tables

**Figure 1 ijerph-19-12831-f001:**
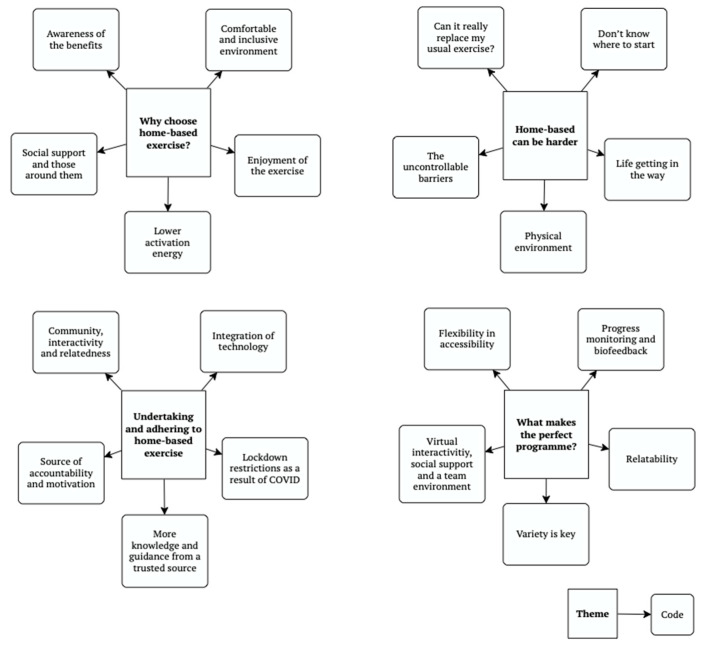
Thematic map of generated codes and themes.

**Table 1 ijerph-19-12831-t001:** Participant demographics (*n* = 19).

Demographic	Participants, *n* (%)
**Sex**	
Male	8 (42%)
Female	11 (58%)
**Age (years)**	
18–24	4 (21%)
25–34	5 (26%)
35–44	2 (11%)
45-54	1 (5%)
55–64	5 (26%)
65+	2 (11%)
**Employment Status**	
Employed	13 (68%)
Retired	6 (32%)
**Primarily working from home**	
Yes	8 (62%)
No	5 (38%)

## Data Availability

The full data are not publicly available in order to maintain participant anonymity.

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
