# Peer review of "“I Was Having My Midlife Fat Crisis”: Exploring the Experiences and Preferences of Home-Based Exercise Programmes for Adults Living with Overweight and Obesity"

_ijerph, 2022, doi:10.3390/ijerph191912831_

Round 1
Reviewer 1 Report
The authors conducted semi-structured interviews virtually to explore 20 overweight participant’s experiences of exercising at home and their preferences for the design of future home-based exercise programs. they concluded that home-based exercise programs for adults living with overweight and obesity are an appropriate option for people wanting to be active. However, programs should be designed in collaboration with people with lived experience.
This is a well-designed and nicely presented study focusing on a highly important topic.
Comments:
Further data on concomitant diseases of the study population should be added.
Medical treatment options could be mentioned, since they can also provide a huge help to the patients.
The manuscript is a bit long, it could be shortened.
Reviewer 2 Report
Overall, I though that this was a well-designed study with very interesting and well described results. However, I do believe the following should be addressed:
(1) Introduction was on the long side - paragraphs 1/3/4 all seemed to be describing the need for the research (ie getting feedback from target population) - could be summarized more succinct
(2) Methods were well described and sound
(3) Results - Table 1, horizontal view of categorical ages was confusing, would be better if verticle
Overall I found it distracting to have the participants names included in the description of study themes/quotations - this seemed to also potentially violate some of the anonymous nature of research. Otherwise, I thought that the choice of highlighted quotes was wellrounded and each added to the paper.
(4) I think the conclusion could most significantly be improved - overall it felt as if it summarized what was included in the results section, and did not add much new. As this is a qualitative study with limited diversity in the sample, I think it is critical to look at the broader literature regarding whether themes they found seem supported - overall, less repetition in conclusion of what was already stated.
(5) Instead of general future directions for research, would also be more specific in terms of how these findings should guide intervention development, and how these findings fit with existing literature
Round 2
Reviewer 2 Report
See attached Word file.
